# Data-Driven Conditional Robust Optimization

**Abhilash Chenreddy**
GERAD & Dept. of Decision Sciences
HEC Montréal
Montréal, Quebec, Canada
abhilash.chenreddy@hec.ca

**Nymisha Bandi**
McGill University
Montréal, Quebec, Canada
nymisha.bandi@mcgill.ca

**Erick Delage**
GERAD & Dept. of Decision Sciences
HEC Montréal
Montréal, Quebec, Canada
erick.delage@hec.ca

## Abstract

In this paper, we study a novel approach for data-driven decision-making under uncertainty in the presence of contextual information. Specifically, we address this problem using a new Conditional Robust Optimization (CRO) paradigm that seeks the solution of a robust optimization problem where the uncertainty set accounts for the most recent side information provided by a set of covariates. We propose an integrated framework that designs the conditional uncertainty set by jointly learning a partition in the covariate data space and simultaneously constructing region specific deep uncertainty sets for the random vector that perturbs the CRO problem. We also provide theoretical guarantees for the coverage provided by conditional uncertainty sets and for the value-at-risk performances obtained using the proposed CRO model. Finally, we use simulated and real world data to illustrate the implementation of our approach and compare it against two non-contextual robust optimization benchmark approaches to demonstrate the value of exploiting contextual information in robust optimization.

## 1 Introduction

In most real world decision problems, the decision maker (DM) faces uncertainty either in the objective function that he aims to optimize, or some of the constraints that he needs to satisfy. Stochastic Programming and Robust Optimization (RO) are the most popular methods for addressing this issue. With the growing availability of data, there has recently been a surge of interest in modeling optimization under uncertainty as contextual optimization problems that seek to leverage rich feature observations to make better decisions [Ban and Rudin, 2019, Bertsimas and Kallus, 2020]. In a simple cost minimization problem, where $\mathcal{X} \subseteq \mathbb{R}^n$ and $c(x, \xi)$ respectively capture the feasible set of actions and a cost that depends on both the action $x$ and a random perturbation vector $\xi \in \mathbb{R}^m$, the "contextual" DM has access to a vector of covariates $\psi \in \mathbb{R}^m$ assumed to be correlated to $\xi$. This DM therefore traditionally wishes to identify an optimal policy, i.e. a functional $\boldsymbol{x} : \mathbb{R}^m \to \mathcal{X}$ that suggests an action in $\mathcal{X}$ adapted to the observed realization of $\psi$, with respect to his expected cost over the joint distribution of $(\psi, \xi)$:

$$\min_{\boldsymbol{x}(\cdot)} \mathbb{E}[c(\boldsymbol{x}(\psi), \xi)]. \tag{1}$$

From a theoretical point of view, one can exploit the interchangeability property (see Theorem 14.60, [Rockafellar and Wets, 2009]) to identify an optimal policy for Problem (1) using the following

conditional stochastic optimization (CSO) problem:

$$\text{(CSO)} \qquad \boldsymbol{x}^*(\psi) \in \underset{x \in \mathcal{X}}{\arg\min}\, \mathbb{E}[c(x, \xi)|\psi]. \qquad (2)$$

While the literature that treats contextual optimization through the CSO problem is rich, much less attention has been given to contextual optimization in the risk averse setting. Namely, one can easily think about replacing the risk neutral expected value operator in problem (2) with a risk measure such as value-at-risk or conditional value-at-risk in order to prevent the DM from being exposed to the possibility of large costs. Moreover, while robust optimization is being used pervasively in disciplines that employ decision models, including chemical, civil, electrical engineering, medicine, and physics (see respectively [Bernardo and Saraiva, 1998, Bendsøe et al., 1994, Mani et al., 2006, Chu et al., 2005, Bertsimas et al., 2007]) to name a few, the question of how to systematically integrate contextual information in this important class of decision models remains to this day unexplored.

In this work, we therefore tackle for the first time the contextual optimization problem from the point of view of robust optimization. Namely, we will consider a contextual DM that wishes to exploit the side information in the design and solution of a robust optimization problem. This naturally gives rise to the following **conditional robust optimization** (CRO) problem

$$\boldsymbol{x}^*(\psi) := \underset{x \in \mathcal{X}}{\arg\min}\, \underset{\xi \in \mathcal{U}(\psi)}{\max}\, c(x, \xi)\,,$$

where $\mathcal{U}(\psi)$ is an uncertainty set designed to contain with high probability the realization of $\xi$ conditionally on observing $\psi$. Our proposed approach will be data-driven in the sense that the design of the CRO problem will make use of historical observations of joint realizations of $\psi$ and $\xi$.

Our contribution can be summarized as follows.

- We propose for the first time a framework for learning from data an uncertainty set for RO that adapts to side information. The "training" of this conditional uncertainty set is done by jointly learning a partition in the covariate data space using deep clustering methods, and simultaneously constructing region specific deep uncertainty sets, using techniques from one-class classification, for the random vector that perturbs the CRO problem.

- We establish theoretical connections between CRO and Contextual Value-at-Risk Optimization (CVO):

$$\underset{\boldsymbol{x}(\cdot)}{\min}\, \text{VaR}_{1-\varepsilon}(c(\boldsymbol{x}(\psi), \xi)), \qquad (3)$$

  where $\text{VaR}_{1-\varepsilon}(Z) := \inf\{t|\mathbb{P}(Z \le t) \ge 1-\varepsilon\}$ refers to the value-at-risk of $1-\varepsilon$ confidence level of $Z$.

- We demonstrate empirically that contextual robust optimization can improve the performance of robust optimization models in a data-driven portfolio optimization problem that employs real world data from the US stock market. In particular, we find that in conditions where side information carries a strong signal about future returns, the risk of the portfolio can be reduced by up to 15%.

The paper is organized as follows. Section 2 surveys related work. Section 3 summarizes the approach discussed in [Goerigk and Kurtz, 2020]. Section 4 presents a Deep Cluster then Classify (DCC) scheme and our Integrated Deep Cluster then Classify (IDCC) scheme to generate conditional uncertainty sets. It also establishes the connections to CVO. Our case study based on real world portfolio optimization is presented in section 5 followed by conclusions in section 6.

## 2 Related work

**Conditional Stochastic Optimization** [Hannah et al., 2010] was possibly the earliest work on CSO, where a kernel density estimation approach is exploited to formulate and solve a CSO problem. [Ban and Rudin, 2019] apply CSO to a newsvendor optimization problem where the performance of linear policies and kernel density estimation is explored and where generalization error can be controlled using regularization. [Kallus and Mao, 2020] studied methods to train forest decision policies for CSO in a way that directly targets the optimization costs. [Ban et al., 2019] use residual tree methods to solve general multi-stage stochastic programs where information about the underlying

uncertainty is available through covariate information. [Kannan et al., 2020a] propose data-driven SAA frameworks for approximating the solution to two-stage stochastic programs with access to a finite number of samples of random variables and concurrently observed covariates. Recently, Lin et al. [2022] has applied a conditional VaR constrained CSO formulation to the newsvendor problem. While most of the related work focuses on an "estimate-then-optimize" approach (see also [Srivastava et al., 2021b] and [Hu et al., 2022]), there have also been recent efforts in designing CSO models using an end-to-end paradigm (see [Elmachtoub and Grigas, 2022] and [Donti et al., 2017]).

**Distributionally robust CSO** One common challenge with the applications of CSO is due to the fact that often there are only a few samples (if any at all) drawn from the conditional distribution of $\xi$ given $\psi$ for each realization of $\psi$ [Hu et al., 2020]. This in turn causes a poor approximation of the true conditional distribution resulting in poor out-of-sample performance. Most proposed solutions to this issue have relied on distributionally robust optimization (DRO). For example, [Bertsimas and Van Parys, 2021], [Bertsimas et al., 2022, Nguyen et al., 2021], and [Srivastava et al., 2021a] all propose DRO approaches that employ distribution sets that are centered at either the estimated conditional distribution or joint empirical distribution of $(\psi, \xi)$. [Kannan et al., 2020b] applies distributionally robust optimization to the residual-based CSO model proposed in [Kannan et al., 2020a]. We finally note that none of these works have considered the problem of conditional DRO where the distributional ambiguity set itself, namely its support or size, depending on contextual information.

**Data-driven Robust Optimization and One-class Classification** There has been a growing set of papers (see [Ohmori, 2021, McCord, 2019, Wang and Jacquillat, 2020]) proposing various frameworks that use both supervised and unsupervised one-class classification techniques in designing the uncertainty sets which are further integrated into the RO problems. Some approaches make use of variance and covariance of historical data [Natarajan et al., 2008] while others [Goerigk and Kurtz, 2020, Wang et al., 2021] have exploited the representative power of deep neural networks to construct compact uncertainty sets Up to this day, none of the data-driven robust optimization approaches have considered accounting for contextual information.

**Deep Clustering Methods** Traditional clustering methods like Gaussian Mixture Models (GMM) and $k$-means clustering rely on the original data representations and suffer from the curse of dimensionality. Recent developments in DNNs led to the learning of high quality representations, especially auto-encoder(AE) and decoder systems are particularly appealing as they are able to learn the representations in a fully unsupervised fashion. Several works like [Chang et al., 2017, Guo et al., 2017, Ji et al., 2017] combine variational AEs and GMMs to perform clustering and non-linearly map the input data into a latent space. Few works like [Fard et al., 2020] try to jointly learn the representations and jointly cluster with $k$-means and learning representations. We modify these algorithms to introduce a probability simplex that interacts with the centroids and also the center of the uncertainty sets.

## 3 The Deep Data-Driven Robust Optimization (DDDRO) Approach

Focusing on a classical robust optimization model, i.e. $\min_{x \in \mathcal{X}} \max_{\xi \in \mathcal{U}} c(x, \xi)$, the authors of [Goerigk and Kurtz, 2020] propose to employ deep learning to characterize the uncertainty set $\mathcal{U}$ in a data-driven environment. In particular, they consider describing the uncertainty set $\mathcal{U}$ in the form:

$$\mathcal{U}(W, R) := \{ \xi \in \mathbb{R}^m : \|f_W(\xi) - \bar{f}_0\| \leq R \}, \tag{4}$$

where $f_W : \mathbb{R}^m \to \mathbb{R}^d$ is a deep neural network, parametrized using $W$, that projects the perturbation vector $\xi$ to a new vector space where the uncertainty set can be more simply defined as a sphere of radius $R$ centered at some $\bar{f}_0$.

Given a dataset $\mathcal{D}_\xi = \{\xi_1, \xi_2 \dots \xi_N\}$, they propose discovering the underlying structure of $\mathcal{U}$ by training the NN using a method found in the one-class classification literature, namely minimizing the empirical centered total variation of the projected data points:

$$\min_W \frac{1}{N} \sum_{i=1}^{N} \|f_W(\xi_i) - \bar{f}_0\|^2, \tag{5}$$

where $\bar{f}_0 := (1/N) \sum_{i \in [N]} f_{W_0}(\xi_i)$ is the center of the projected points under some initial random choice of $f_{W_0}$. Once the network is trained, they calibrate the radius $R$ of $\mathcal{U}$ in order to reach a targeted coverage $1 - \epsilon$ of the data set.

In terms of NN architecture, they favor a special class of fully connected neural networks of depth $L$:

$$f_W(c) = \sigma^L(W^L \sigma^{L-1}(W^{L-1} \dots \sigma^1(W^1(c)) \dots)) \tag{6}$$

where each $W^\ell$ captures a linear projection while each $\sigma^\ell$ captures a term-wise piecewise linear activation function (e.g. ReLU, Hardtanh, or hard sigmoid):

$$\sigma_j^\ell(w_j) = a_k^\ell w_j + b_k^\ell \ \text{ if } \ \underline{\alpha}_k^\ell \le w_j \le \overline{\alpha}_k^\ell, \quad k = 1, \dots, K$$

with $\{a_k^\ell, b_k^\ell, \underline{\alpha}_k^\ell, \overline{\alpha}_k^\ell\}_{k=1}^K$ as the parameters that identifies each of the $K$ affine pieces.

The motivation for such an architecture comes from the proposed solution scheme for the RO problem, which relies on a constraint generation approach (See Algorithm 3, 4 in Appendix). This scheme relies on progressively adding scenarios to a reduced set $\mathcal{U}' \subseteq \mathcal{U}$ until the worst-case cost of the solution under $\mathcal{U}'$ is the same as under $\mathcal{U}$. Numerically, a critical step consists in identifying the worst-case realization in $\mathcal{U}$, which is shown to reduce to a mixed-integer linear program when $c(x, \xi)$ is linear in $\xi$ under the selected NN architecture due to the following representation of $\mathcal{U}(W, R)$:

$$\mathcal{U}(W, R) = \left\{ \xi \left| \begin{array}{l} \exists u \in \{0, 1\}^{d \times K \times L}, \ \zeta \in \mathbb{R}^{d \times L}, \ \phi \in \mathbb{R}^{d \times L} \\ \sum_{k=1}^K u_j^{k,\ell} = 1, \ \forall j, \ell \\ \phi^1 = W^1 \xi \\ \zeta_j^\ell = \sum_{k=1}^K u_j^{k,\ell} a_k^\ell \phi_j^\ell + \sum_{k=1}^K u_j^{k,\ell} b_k^\ell, \ \forall j, \ell \\ \phi^\ell = W^\ell \zeta^{\ell-1}, \ \forall \ell \ge 2 \\ \sum_{k=1}^K u_j^{k,\ell} \underline{\alpha}_k^\ell \le \phi_j^\ell \le \sum_{k=1}^K u_j^{k,\ell} \overline{\alpha}_k^\ell, \ \forall j, \ell \\ \|\zeta^L - \bar{f}_0\| \le R \end{array} \right. \right\}, \tag{7}$$

where we assume for simplicity that each layer of the deep neural network has $d$ neurons and $\phi^\ell$ is the output at $l$-th layer of the neural network. We refer interested readers to [Goerigk and Kurtz, 2020] for more details.

## 4 Deep Data-driven Conditional Robust Optimization

Let $(\psi, \xi)$ be a pair of random vectors defining respectively the side-information and random perturbation vectors of a contextual optimization problem. We can call our dataset $\mathcal{D}_{\psi\xi} := \{(\psi_1, \xi_1), \dots, (\psi_N, \xi_N)\}$. Our objective is to train a data-driven conditional uncertainty set $\mathcal{U}(\psi)$ that will lead to robust solutions that are adapted to the type of perturbance that is experienced when $\psi$ is observed. In this section, we propose two algorithms, namely the Deep cluster then classify (DCC) and the Integrated Deep cluster then classify (IDCC), to do so, and propose a calibration procedure that offers some guarantees with respect to a contextual value-at-risk problem.

### 4.1 The Deep "Cluster then Classify" (DCC) Approach

A direct extension of G&K's DDDRO approach in section 3 consists in reducing the side-information $\psi$ to a set of $K$ different clusters, which provides states of the environment in which one wishes to design customized data-driven uncertainty sets. Mathematically, $\mathcal{U}(\psi) := \mathcal{U}_{a(\psi)}$, where $a : \mathbb{R}^m \to [K]$, is a trained $K$-class cluster assignment function for $\psi$, and each $\mathcal{U}_k$, for $k = 1, \dots, K$, is an uncertainty sets for $\xi$ that is trained and sized using the procedure described in section 3 with the dataset $\mathcal{D}_\xi^k := \cup_{(\psi, \xi) \in \mathcal{D}_{\psi\xi} : a(\psi) = k} \{\xi\}$. This process implicitly involves multiple sequential steps of training deep neural networks. Following [Moradi Fard et al., 2020], when performing deep $K$-mean clustering to obtain $a(\psi)$, training can take the form of Algorithm 5, where the deep $K$-means algorithm trains simultaneously a representation $g_{V_E} : \mathbb{R}^m \to \mathbb{R}^d$, using an encoder and $g_{V_D} : \mathbb{R}^d \to \mathbb{R}^m$, using a decoder network, and a K-mean classifier $\bar{a}^\theta(\phi) := \operatorname{argmin}_{k \in [K]} \|\phi - \theta^k\|_2$ by minimizing, using stochastic gradient descent in a coordinate descent scheme, a trade-off (using $\alpha_K$) between reconstruction error and the within cluster centered total variation in the encoded space:

$$\mathcal{L}^1(V, \theta) := (1 - \alpha_K) \frac{1}{N} \sum_{i=1}^N \|g_{V_D}(g_{V_E}(\psi_i)) - \psi_i\|^2 + \alpha_K \frac{1}{N} \sum_{i=1}^N \|g_{V_E}(\psi_i) - \theta^{a(\psi_i)}\|^2, \tag{8}$$

where $a(\psi) := \bar{a}^\theta(g_{V_E}(\psi))$. To solve this problem, we iterate between improving $V := (V_E, V_D)$ while keeping $\theta$ fixed, and improving $\theta$ while preserving $V$ fixed.

Once the $K$-mean and one-class classifiers are trained, we correct for a deficiency of DDDRO approach, which assumes wrongfully that the projected $f_{W^k}(\xi)$ are normalized for each $\mathcal{D}_\xi^k$. Namely, we replace $\mathcal{U}(W, R)$ with a set that employs an ellipsoid in the projected space according to the statistics of $\mathcal{D}_\xi^k$:

$$\mathcal{U}(W^k, R^k, \mathcal{S}^k) := \{\xi \in \mathbb{R}^m : \|\Sigma_f^{k-1/2}(f_{W^k}(\xi) - \mu_f^k)\| \le R^k\}, \tag{9}$$

where $\mathcal{S}^k$ is short for $(\mu_f^k, \Sigma_f^k)$ with

$$\mu_f^k := |\mathcal{D}_\xi^k|^{-1} \sum_{\xi \in \mathcal{D}_\xi^k} f_{W_0^k}(\xi) \text{ and } \Sigma_f^k := |\mathcal{D}_\xi^k|^{-1} \sum_{\xi \in \mathcal{D}_\xi^k} (f_{W^k}(\xi) - \mu^k)(f_{W^k}(\xi) - \mu^k)^T.$$

The calibration of each $R^k$ can finally be done using the same procedure as in [Goerigk and Kurtz, 2020]but using the reduced dataset $\mathcal{D}_\xi^k$.

## 4.2 The Integrated Deep Cluster-Classify (IDCC) Approach

While the simplicity of the approach presented in section 4.1 makes it appealing, we identify two important weaknesses. First, by separating the training into multiple steps, it omits tackling the conditional uncertainty set learning problem as a whole. Namely, that low total variation in the $\psi$ space (or a projection of it) does not necessarily imply that low total variation can easily be achieved in a projection of the $\xi$ space. Second, it is unclear how to adapt the approach to a context where a clear separation of the clusters is impossible and where the notion of partial membership to a cluster is more appropriate.

To address the first problem, we propose an integrated framework for performing deep clustering and deep uncertainty set design jointly. Namely, we propose to optimize all of $V$, $\theta$, and $\{W^k\}_{k=1}^K$ jointly using a loss function that trades-off between the objectives used for clustering and each of the $K$ versions of one-class classifiers. We also tackle the issue of hard assignments by training a parameterized random assignment policy $\pi : \mathbb{R}^m \to \Delta_K$, where $\Delta_K$ is the probability simplex in $\mathbb{R}^K$, and $\theta$ the parameters that define the policy space. In the context of employing a soft version of deep $K$-means [Fard et al., 2020]; this random assignment policy takes the form of $\pi(\psi) := \bar{\pi}^\theta(g_V(\psi))$, where

$$\bar{\pi}_k^\theta(\psi) := \frac{\exp\{-\beta\|g_V(\psi) - \theta^k\|^2\}}{\sum_{k'=1}^K \exp\{-\beta\|g_V(\psi) - \theta^{k'}\|^2\}} \tag{10}$$

With these adjustments, our proposed loss function takes the form of:

$$\begin{aligned}
\mathcal{L}_\alpha^3(V, \theta, \{W^k\}_{k=1}^K) := &\alpha_S\Big((1 - \alpha_K)\mathbb{E}_\mathcal{D}^\pi[\|g_{V_D}(g_{V_E}(\psi_i)) - \psi_i\|^2] \\
&+ \alpha_K \mathbb{E}_\mathcal{D}^\pi[\text{TotalVar}_\mathcal{D}^\pi(g_{V_E}(\psi), \theta^{\tilde{a}(\psi)} | \tilde{a}(\psi))]\Big) \\
&+ (1 - \alpha_S)\frac{1}{K} \sum_{k=1}^K \min_{\vartheta^k} \text{TotalVar}_\mathcal{D}^\pi(f_{W^k}(\xi), \vartheta^k | \tilde{a}(\psi) = k), \tag{11}
\end{aligned}$$

where $\tilde{a}(\psi) \sim \bar{\pi}^\theta(g_{V_E}(\psi))$ is the randomized assignment based on $\psi$, $\text{TotalVar}_\mathcal{D}^\pi(\phi, \theta | \tilde{a}(\psi)) := \sum_{j=1}^d \mathbb{E}_\mathcal{D}^\pi[(\phi_j - \theta_j)^2 | \tilde{a}(\psi)]$ is the conditional centered total variation of given $\tilde{a}(\psi)$. In fact, all statistics are measured using the empirical distribution expressed in $\mathcal{D}_{\psi\xi}$ and the conditional distribution produced by the randomized assignment policy $\bar{\pi}^\theta(g_V(\psi))$, i.e. $\mathbb{P}_\mathcal{D}^\pi((\psi, \xi, \tilde{a}) \in \mathcal{E}) = (1/N)\sum_{i=1}^N \sum_{k=1}^K \mathbf{1}\{(\psi_i, \xi_i, k) \in \mathcal{E}\}\bar{\pi}_k^\theta(g_V(\psi_i))$. The explicit form of equation (11) can be found in Appendix B.1.

Overall, $\mathcal{L}_\alpha^3$ trades off (using $\alpha_S$) between the reconstruction error of the encoder-decoder networks on $\xi$, the expected recognizability of the $K$ clusters, i.e. the fact that the observed features $g_{V_E}(\psi)$ form distinct clusters of points, and the average compactness of the produced conditional uncertainty

sets. In particular, as $\alpha_S \to 1$, we can expect the minimizer of $\mathcal{L}_\alpha^3$ to converge to the minimizer of the cluster and classify approach. At the other end of the spectrum, when $\alpha_S \to 0$, the model will produce more self contained conditional uncertainty sets but at the price of less distinguishable clusters (in terms of $\psi$) that might poorly exploit the side-information. Algorithm 1 presents our proposed training scheme for the IDCC approach.

Given that we employ a random assignment policy, we propose replacing the deterministic CRO problem with its randomized version:

$$\tilde{\boldsymbol{x}}^*(\psi) \in \underset{x \in \mathcal{X}}{\operatorname{argmin}} \max_{\xi \in \tilde{\mathcal{U}}(\psi)} c(x, \xi),$$

where $\tilde{\mathcal{U}}(\psi) := \mathcal{U}(W^{\tilde{a}(\psi)}, R^{\tilde{a}(\psi)}, \mathcal{S}^{\tilde{a}(\psi)})^1$ is a random uncertainty set, and where we express the fact that conditionally on $\psi$, $\tilde{\boldsymbol{x}}(\psi)$ is a random policy that depends on the realization of $\tilde{a}$. Given the randomness of $\tilde{\mathcal{U}}(\psi)$, one needs to be more careful in defining a calibration scheme for each $R^k$. Our proposed scheme is motivated by the following Lemma, which proof can be found in appendix A.

**Lemma 4.1.** *Let the random uncertainty set $\tilde{\mathcal{U}}(\psi)$ satisfy:*

$$\mathbb{P}_{\mathcal{D}}^\pi(\xi \in \tilde{\mathcal{U}}(\psi) | \tilde{a}(\psi) = k) \geq 1 - \epsilon, \forall k, \tag{12}$$

*then it satisfies:*

$$\mathbb{P}_{\mathcal{D}}^\pi(\xi \in \tilde{\mathcal{U}}(\psi)) \geq 1 - \epsilon. \tag{13}$$

In particular, this lemma suggests calibrating each $R^k$ using the bisection to solve:

$$\inf \left\{ R \, \middle| \, \frac{\sum_{i=1}^N \mathbf{1}\{\xi_i \in \mathcal{U}(W^k, R, \mathcal{S}^k)\} \bar{\pi}_k^\theta(g_{V_E}(\psi_i))}{\sum_{i=1}^N \bar{\pi}_k^\theta(g_{V_E}(\psi_i))} \geq 1 - \epsilon \right\}, \tag{14}$$

given that the resulting $R^k$ are the smallest that satisfy (12).

---

**Algorithm 1** Integrated deep cluster-classify with deep $K$-means

---

Input: Data-set $\mathcal{D}_{\psi\xi}$, Number of clusters $K$, hyper-parameters $\alpha_K, \alpha_S, \beta$
Randomly initialize $\theta_0$, $V_0$, and $W_0$
Let $\pi_0 := \bar{\pi}^{\theta_0}(g_{V_{E0}}(\psi))$ and $W_0^k := W_0$ for all $k$'s
Set $t := 0$
**repeat**
    Set $t := t + 1$.
    Update $\theta_t^k := \mathbb{E}_{\mathcal{D}}^\pi[g_{V_{Et-1}}(\psi) | \tilde{a}(\psi) = k]$ using $\pi_{t-1}$
    Update $(V_t, \{W_t^k\}_{k=1}^K)$ using gradient descent on (11) with $\theta_t$
    Get $\pi_t := \bar{\pi}^{\theta_t}(g_{V_{Et}}(\psi))$
**until** $t \geq T$ or convergence
Let $\pi(\cdot) := \pi_t(\cdot)$ and $W^k := W_t^k$ for all $k$'s
**for** $k = 1, \dots, K$ **do**
    Calibrate $R^k$ using (14)
    Let $\mathcal{U}^k := \mathcal{U}(W^k, R^k, \mathcal{S}^k)$
**end for**
Return $\pi(\cdot)$ and $\{\mathcal{U}^k\}_{k=1}^K$

---

### 4.3 Connections to Contextual Value-at-Risk Optimization

In the previous subsections, we proposed two different schemes to produce a possibly randomized uncertainty set $\tilde{\mathcal{U}}(\psi)$ that can be employed in a randomized CRO problem.[2] We also proposed a

---

[1]Here, $\mathcal{S}^k$ refers to $(\bar{f}_{W^k | \tilde{a}(\psi_i) = k}^{\theta, V}, \bar{\Sigma}_{W_k | \tilde{a}(\psi) = k}^{\theta, V})$ with

$$\bar{\Sigma}_{W_k | \tilde{a}(\psi) = k}^{\theta, V} := \sum_{i=1}^N \frac{\bar{\pi}_k^\theta(g_{V_E}(\psi_i))}{\sum_{i=1}^N \bar{\pi}_k^\theta(g_{V_E}(\psi_i))} \cdot (f_{W^k}(\xi_i) - \bar{f}_{W^k | \tilde{a}(\psi_i) = k}^{\theta, V})(f_{W^k}(\xi_i) - \bar{f}_{W^k | \tilde{a}(\psi_i) = k}^{\theta, V})^T$$

.

[2]Note that in the case of section 4.1, the conditional uncertainty set is deterministic thus reducing the randomized version of CRO to a pure CRO problem

scheme for radii calibration so that they would satisfy the coverage property in (13). Hence, one can derive the following connection between conditional robust optimization and the CVO problem (1). The proof is pushed to Appendix A.

**Lemma 4.2.** *When $\tilde{\mathcal{U}}$ satisfies (13), the random policy $\tilde{\boldsymbol{x}}(\cdot)$ to the randomized CRO problem together with*

$$v^* := esssup_{\mathcal{D}}^{\pi} \min_{x \in \mathcal{X}} \max_{\xi \in \tilde{\mathcal{U}}(\psi)} c(x, \xi)$$

*provide a conservative approximate solution to the CVO problem under the empirical measure $\mathbb{P}_{\mathcal{D}}^{\pi}$. Namely,*

$$VaR_{1-\epsilon}^{\mathcal{D},\pi}(c(\tilde{\boldsymbol{x}}(\psi), \xi)) \leq v^*.$$

*In particular, in the case of the proposed DCC and IDCC approaches we have that*

$$v^* = \max_{k \in [K]} \min_{x \in \mathcal{X}} \max_{\xi \in \mathcal{U}(W^k, R^k, \mathcal{S}^k)} c(x, \xi).$$

As the robust optimization paradigm traditionally aims at offering statistical guarantees on the out-of-sample performance of the prescribed solutions, we describe below how a bootstrap method can be used to estimate the radii $R^k$'s.

**Remark 4.1.** *Using bootstrapping methods, we can get a conservative approximation of each $R_k$ as:*

$$\tilde{R}_k := \inf \left\{ R \left| \mathbb{P}_{\tilde{\mathcal{D}}} \left( \sum_{i=1}^{N} \frac{\bar{\pi}_k^{\theta}(g_{V_E}(\psi_i)}{\sum_{i=1}^{N} \bar{\pi}_k^{\theta}(g_{V_E}(\psi_i)} \mathbf{1}\{\xi_i \in \mathcal{U}(W^k, R, \mathcal{S}^k)\} \geq 1 - \epsilon \right) \geq 1 - \delta \right\} \right.$$

*where $\mathbb{P}_{\tilde{\mathcal{D}}}$ measures the probability when resampling a new dataset of size $N$ with replacement from $\mathcal{D}_{\psi\xi}$. When $N$ is large enough and assuming that each data point is drawn i.i.d. according to some unknown probability measure $\mathbb{P}$, we asymptotically get the guarantee that $\mathbb{P}(\xi \in \tilde{\mathcal{U}}(\psi)) \geq 1 - \epsilon$ with probability higher than approximately $1 - K\delta$.*

## 5 Experiments

In this section, we illustrate the coverage aspect of the IDCC approach using simulated data. We will further demonstrate the advantage of the CRO problem using a standard risk minimizing portfolio optimization problem. We compare the performance of IDCC with that of DCC, DDDRO (with ellipsoidal correction in (9)), and the classical ellipsoidal uncertainty approach (i.e. DCC with $K = 1$ and $f_{W^1}(\xi) := \xi$). The IDCC and DCC methods incorporate the covariate information whereas DDDRO and ellipsoid approaches ignore this information. The neural network architecture and other modeling information are available in Appendix B. The code can be found on github[3]. Our code uses the Pytorch implementation from [Goerigk and Kurtz, 2020], which is available online[4].

### 5.1 Conditional uncertainty set illustration using simulated data

For ease of illustration, we consider a simulation environment where $[\psi^T \ \xi^T] \in \mathbb{R}^4$ is a random vector whose distribution is an equal-weighted mixture of two 4-d multivariate normal distributions. We consider $N = 500$ and train IDCC (with $K = 2$), DDDRO, and the ellipsoid and calibrate the uncertainty sets for a probability coverage of 90%, 99% (i.e. $\epsilon \in \{1\%, \ 10\%\}$). As a result, DDDRO and IDCC, which use deep neural networks, identify non-convex uncertainty sets, whose convex hulls are presented in Figure 1 together with the calibrated ellipsoid. The figure also presents the conditional distribution of $\xi$ according to $\mathbb{P}_{\mathcal{D}}^{\pi}(\cdot|\tilde{a}(\psi) = k)$, using IDCC's randomized assignment, and the training dataset. One can remark that the conditional sets produced by IDCC exploit the side information by concentrating the uncertainty set on the region that has the most mass according to $\mathbb{P}_{\mathcal{D}}^{\pi}(\cdot|\tilde{a}(\psi) = k)$ thus leading to a less conservative RO problem then DDDRO and the ellipsoid, which are oblivious to $\psi$. In fact, it appears to have successfully learned to at least partially recognize the mixture membership using $\psi$ and exploit this information to adapt the uncertainty set.

---

[3] https://anonymous.4open.science/r/Data-Driven-Conditional-Robust-Optimization-E160/
[4] https://github.com/goerigk/RO-DNN

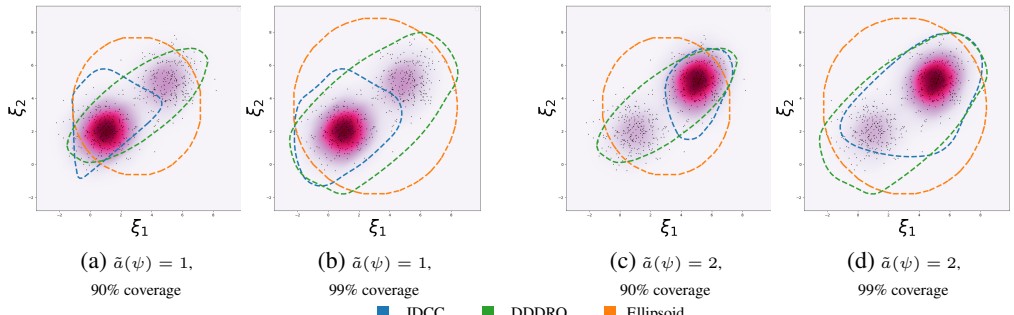

$$\begin{array}{cccc}
\text{(a) } \tilde{a}(\psi) = 1, & \text{(b) } \tilde{a}(\psi) = 1, & \text{(c) } \tilde{a}(\psi) = 2, & \text{(d) } \tilde{a}(\psi) = 2, \\
\text{90\% coverage} & \text{99\% coverage} & \text{90\% coverage} & \text{99\% coverage}
\end{array}$$

■ IDCC ■ DDDRO ■ Ellipsoid

Figure 1: Convex hull of trained uncertainty sets for two levels of coverage and with a conditional uncertainty set for IDCC that exploits two clusters. The heatmap represents the conditional distribution of $\xi$ according to $\mathbb{P}^\pi_{\mathcal{D}}(\cdot|\tilde{a}(\psi) = k)$. The cloud of points represents the training dataset.

## 5.2  Robust portfolio optimization

We further investigate the empirical out-of-sample performance of the proposed uncertainty sets on a classical robust portfolio optimization problem. Namely, we consider a situation where an investor is trying to minimize the worst-case return based on an uncertainty set that provides $1 - \epsilon$ probabilistic coverage of the uncertain future return vector. In particular, given that $x$ captures a vector of investment in $n = m$ different assets whose return are captured using $\xi$, we let $c(x, \xi) := -\xi^\mathsf{T} x$ to capture the return on investment, and let $\mathcal{X} := \{x \in \mathbb{R}^n | \sum_{i=1}^n x_i = 1, \ x \geq 0\}$ to capture the need to invest one unit of wealth among the available assets. Following Lemma 4.2, this model can in turn be interpreted as conservatively approximating a $\min_{x \in \mathcal{X}} \text{VaR}_{1-\epsilon}(\xi^\mathsf{T} x)$, where the objective is a risk averse value-at-risk metric.

**Dataset** Our experiments make use of historical data from the US stock market. We collect the adjusted daily closing prices for 70 stocks (as used in [Xu and Cohen, 2018]) coming from 8 different sectors from January 1, 2012, to December 31, 2020, using the Y!Finance's API. Each year has 252 data points and we compute the percentage gain/loss w.r.t the previous day to create our dataset for $\xi$. As for side information, we use the trading volume of individual stocks and other market indices[5] over the same period as covariates. Our algorithm gives the flexibility to use any number of such metrics as contextual information. Given the time series nature of the data, at a given instance, we use 3 years of data to train and the following year as validation to pick the hyperparameters of our model such as learning rate, weight decay, and the optimal number of clusters. We then retrain the model using the 4 years of data to build the final model. Upon calibrating the uncertainty set, we use it to solve the robust portfolio optimization problem. We then apply this policy to the next 1 year's of data and compute the performance metric, namely Value at risk (VaR) for different confidence levels to compare the performances. VaR quantifies the level of risk of a portfolio over a specified time frame. Here, it gives an estimate of the maximum % loss the decision maker can incur over a period of 1 year when he uses the policy from the RO model. Intuitively, lower the VaR, less riskier is the generated policy. Many financial institutions use VaR to determine the amount of collateral needed when trading financial products so lowering VaR for high confidence levels is crucial.

**Experiment Design** To test for the robustness of the IDCC algorithm, we experiment on various randomly sampled stock combinations across different time periods. We randomly sampled a subset of 15 stocks in a time window and repeated the experiment for 10 runs on 3 moving time frames. We used learning rate $= 0.01$, $\alpha_K = 0.5, \alpha_S = 0.5, \beta = 0.1$ for all the experiments. We use a cold start K-means approach to determine K for each run. We do this across all these experiments as it will be computationally expensive to tune the parameters through grid search for each run and also our intention is to show the learning capability of our algorithm even with minimal tuning. The parameter tuning and implementation details can be found in appendix B.3.

---

[5]Volatility Index (VIX), 10-year Treasury Yield Index (TNX), Oil Index (CL=F), S&P 500 (GSPC), Global Income & Currency Fund (XGCFX), Dow Jones Index (DJI)

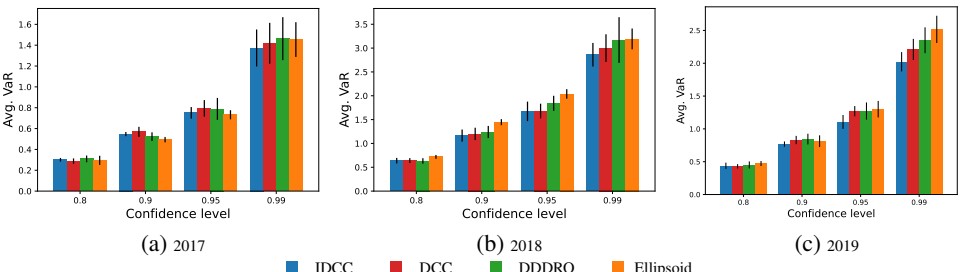

Figure 2: Avg. VaR across portfolio simulations. Error bars report 95% CI.

**Results** Fig. 2 shows the avg. VaR across the runs at different confidence levels. It is evident that IDCC generally performs better than the baseline models. This difference is especially noticeable at a higher confidence level and vanishes as we move to lower confidence levels. Table 1 provides more details by comparing the overall and conditional cluster level VaR with the baseline models. Specifically, in each run, we identify each cluster as either the "majority" or "minority" cluster depending on its frequency and report averages of VaR (among the 10 runs) for each of these labels. The average frequencies for each label are also reported in the table. In particular, one can observe that the improvement on average overall VaR can reach up to ∼15% (see in 2019 at a 0.99 confidence level). This advantage is even more clearly visible when we look at the individual cluster-level conditional VaR. For instance, in the year 2018 for the 0.99 confidence level, the majority cluster (∼68% data) provides an improvement of 19% and an overall improvement of 9% compared to the second best baseline model. A similar pattern is observed for the year 2019 as well. In the year 2017, the overall performance of IDCC is close and for some confidence levels slightly above the baseline models. However, we see that the majority cluster (∼80% data) is performing better than the baseline models while the minority cluster has a slightly higher risk. We attribute this loss in performance to the fact that the minority clusters are much less frequent (∼20% data) and therefore have fewer data available to properly learn its conditional uncertainty set. This large difference in frequencies might also indicate that the side information does not have a strong signal for the behavior of the returns during this period of time.

| | | 2017 | | | | 2018 | | | | 2019 | | | |
|---|---|---|---|---|---|---|---|---|---|---|---|---|---|
| | Conf. $1 - \epsilon$ | 0.8 | 0.9 | 0.95 | 0.99 | 0.8 | 0.9 | 0.95 | 0.99 | 0.8 | 0.9 | 0.95 | 0.99 |
| | IDCC | 0.30 | 0.55 | 0.75 | **1.37** | 0.64 | **1.16** | **1.67** | **2.86** | **0.44** | **0.77** | **1.11** | **2.02** |
| Overall | DDDRO | 0.31 | 0.52 | 0.79 | 1.46 | **0.63** | 1.24 | 1.84 | 3.17 | 0.45 | 0.84 | 1.27 | 2.35 |
| | Ellipsoid | 0.30 | **0.49** | 0.75 | 1.45 | 0.72 | 1.45 | 2.04 | 3.19 | 0.47 | 0.81 | 1.30 | 2.52 |
| Cond. on | Cluster Freq. | | 80% | | | | 68% | | | | 59% | | |
| Majority | IDCC | 0.31 | 0.52 | **0.71** | **1.30** | 0.57 | 1.08 | 1.50 | 2.62 | 0.44 | 0.75 | 1.17 | **1.88** |
| Cluster | DDDRO | 0.31 | 0.52 | 0.74 | 1.35 | 0.59 | 1.15 | 1.63 | 3.23 | 0.45 | 0.85 | 1.31 | 2.06 |
| | Ellipsoid | 0.32 | 0.52 | 0.74 | 1.41 | 0.69 | 1.29 | 1.92 | 3.08 | 0.47 | 0.85 | 1.25 | 2.31 |
| Cond. on | Cluster Freq. | | 20% | | | | 32% | | | | 41% | | |
| Minority | IDCC | 0.30 | 0.61 | 0.77 | 1.43 | **0.96** | **1.57** | 2.05 | **3.13** | **0.48** | 0.82 | **1.15** | **2.22** |
| Cluster | DDDRO | 0.30 | 0.56 | 0.84 | 1.39 | 1.00 | 1.66 | **2.04** | 3.30 | 0.49 | 0.84 | 1.40 | 2.39 |
| | Ellipsoid | **0.28** | **0.47** | **0.69** | **1.13** | 1.17 | 1.80 | 2.43 | 3.43 | 0.49 | 0.82 | 1.38 | 2.57 |

Table 1: Comparison of average value-at-risk (over 10 runs) for different levels of probability coverage. Both the overall VaR and conditional VaR given the membership to the majority/minority clusters are presented.

## 6   Conclusion and Future Work

In this work, we introduced a new approach, Conditional Robust Optimization, for solving contextual optimization problems in a risk averse setting. We proposed a novel integrated approach to design uncertainty sets that adapt to revealed covariate information. We identified connections to contextual value-at-risk optimization and showed empirically that our method reduces the out-of-sample VaR

considerably compared to non-contextual RO schemes when the level of protection needed is high. As future work, we find that it should be interesting to integrate data-driven conditional uncertainty sets in the context of multi-stage robust optimization models. Given that clustering techniques are often prone to learning correlations from the data that do not reflect true causal relations, so there might be a need to integrate causal inference methods into our approach. One might also be concerned regarding fairness considerations in contexts where side information might allow to treat of a certain class of individuals differently from others. This last issue might be addressed by adding fairness consideration in our integrated loss function.

## Acknowledgement

The authors gratefully acknowledge support from the Institut de Valorisation des Données (IVADO) and from the Canadian Natural Sciences and Engineering Research Council [Grant RGPIN-2016-05208 and 492997-2016].

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
