# A Proofs

As mentioned in sections 3 and 4, our dataset $\mathcal{D}$ contains the random perturbation vectors $\xi$ and side information $\psi$. $\tilde{\mathcal{U}}(\psi)$ represents the conditional uncertainty set that satisfies the following properties.

**Lemma 4.1.** *Let the random uncertainty set $\tilde{\mathcal{U}}(\psi)$ satisfy:*

$$\mathbb{P}_{\mathcal{D}}^{\pi}(\xi \in \tilde{\mathcal{U}}(\psi)|\tilde{a}(\psi) = k) \geq 1 - \epsilon, \ \forall k \tag{15}$$

*then it satisfies:*

$$\mathbb{P}_{\mathcal{D}}^{\pi}(\xi \in \tilde{\mathcal{U}}(\psi)) \geq 1 - \epsilon. \tag{16}$$

*Proof.* The claim follows from:

$$\mathbb{P}_{\mathcal{D}}^{\pi}(\xi \in \tilde{\mathcal{U}}(\psi)) = \sum_{k=1}^{K} \mathbb{P}_{\mathcal{D}}^{\pi}(\xi \in \tilde{\mathcal{U}}(\psi)|\tilde{a}(\psi) = k)\mathbb{P}_{\mathcal{D}}^{\pi}(\tilde{a}(\psi) = k)$$

$$\geq \sum_{k}(1 - \epsilon)\mathbb{P}_{\mathcal{D}}^{\pi}(\tilde{a}(\psi) = k) = 1 - \epsilon.$$

$\square$

**Lemma 4.2.** *When $\tilde{\mathcal{U}}$ satisfies* (13)*, the random policy $\tilde{\boldsymbol{x}}(\cdot)$ to the randomized CRO problem together with*

$$v^* := esssup_{\mathcal{D}}^{\pi} \min_{x \in \mathcal{X}} \max_{\xi \in \tilde{\mathcal{U}}(\psi)} c(x, \xi)$$

*provide a conservative approximate solution to the CVO problem under the empirical measure $\mathbb{P}_{\mathcal{D}}^{\pi}$. Namely,*

$$VaR_{1-\epsilon}^{\mathcal{D},\pi}(c(\tilde{\boldsymbol{x}}(\psi), \xi)) \leq v^*.$$

*In particular, in the case of the DCC and IDCC approaches we have that*

$$v^* = \max_{k \in [K]} \min_{x \in \mathcal{X}} \max_{\xi \in \mathcal{U}(W^k, R^k, \mathcal{S}^k)} c(x, \xi).$$

*Proof.* First, by definition of $\tilde{\boldsymbol{x}}(\cdot)$ and $v^*$, we have that when $\xi \in \tilde{\mathcal{U}}(\psi)$:

$$c(\tilde{\boldsymbol{x}}(\psi), \xi) \leq \max_{\xi \in \tilde{\mathcal{U}}(\psi)} c(\tilde{\boldsymbol{x}}(\psi), \xi) = \min_{x \in \mathcal{X}} \max_{\xi \in \tilde{\mathcal{U}}(\psi)} c(x, \xi) \leq v^*.$$

Hence, we must have that:

$$\mathbb{P}_{\mathcal{D}}^{\pi}(c(\tilde{\boldsymbol{x}}(\psi), \xi) \leq v^*) \geq \mathbb{P}_{\mathcal{D}}^{\pi}(c(\tilde{\boldsymbol{x}}(\psi), \xi) \leq v^*|\xi \in \tilde{\mathcal{U}}(\psi))\mathbb{P}_{\mathcal{D}}^{\pi}(\xi \in \tilde{\mathcal{U}}(\psi))$$

$$\geq 1 \cdot (1 - \epsilon).$$

We thus obtain our result based on the following argument:

$$VaR_{1-\varepsilon}^{\mathcal{D},\pi}(c(\tilde{\boldsymbol{x}}(\psi), \xi)) := \inf\{t|\mathbb{P}_{\mathcal{D}}^{\pi}(c(\tilde{\boldsymbol{x}}(\psi), \xi) \leq t) \geq 1 - \epsilon\} \leq v^*.$$

In the case of the DCC and IDCC approaches we have that

$$v^* = \max_{k \in [K]} \min_{x \in \mathcal{X}} \max_{\xi \in \mathcal{U}(W^k, R^k)} c(x, \xi),$$

since $\tilde{\mathcal{U}}(\psi)$ is supported on $\{\mathcal{U}(W^k, R^k, \mathcal{D}_k^{\pi})\}_{k=1}^{K}$.

$\square$

# B Deep learning implementation of IDCC approach

## B.1 Loss function

Mathematically, the conditional total variation loss function (11) can be explicitly written as:

$$\mathcal{L}_{\alpha}^{3}(V, \theta, \{W^k\}_{k=1}^{K}) := (1 - \alpha_S)\frac{1}{K}\sum_{k=1}^{K}\sum_{i=1}^{N}\frac{\bar{\pi}_k^{\theta}(g_{V_E}(\psi_i))}{\sum_{i=1}^{N}\bar{\pi}_k^{\theta}(g_{V_E}(\psi_i))}\|f_{W^k}(\xi_i) - \bar{f}_{W^k|\tilde{a}(\psi_i)=k}^{\theta,V}\|^2$$

$$+ \alpha_S\left((1 - \alpha_K)\frac{1}{N}\sum_{i=1}^{N}\|g_{V_D}(g_{V_E}(\psi_i)) - \psi_i\|^2 + \alpha_K\frac{1}{N}\sum_{i=1}^{N}\sum_{k=1}^{K}\bar{\pi}_k^{\theta}(g_{V_E}(\psi_i))\|g_{V_E}(\psi_i) - \theta^k\|^2\right)$$

$$\tag{17}$$

where

$$\bar{f}_{W^k|\tilde{a}(\psi_i)=k}^{\theta,V} := \sum_{i=1}^{N}\frac{\bar{\pi}_k^{\theta}(g_{V_E}(\psi_i))}{\sum_{i=1}^{N}\bar{\pi}_k^{\theta}(g_{V_E}(\psi_i))}f_{W^k}(\xi_i).$$

## B.2 Architecture

The joint loss minimization task is performed using the following network architecture which has 2 parallel networks training simultaneously. The first network($g_V := (g_{V_E}, g_{V_D})$) takes the side information($\psi$) as the input and generates a randomized assignment $\tilde{a}(\psi) \sim \bar{\pi}^\theta(g_{V_E}(\psi))$. The second network($\{f_{W^k}\}_{k=1}^K$) takes the random perturbation vector($\xi$) and $\tilde{a}(\psi)$ as the input to generate $W^{\tilde{a}(\psi)}, \mathcal{S}^{\tilde{a}(\psi)6}$ which are subsequently used to design the uncertainty set $\tilde{\mathcal{U}}(\psi) := \mathcal{U}(W^{\tilde{a}(\psi)}, R^{\tilde{a}(\psi)}, \mathcal{S}^{\tilde{a}(\psi)})$.

$g_V$ is an auto-encoder(AE) network which generates the assignment vector $\tilde{a}(\psi)$. They are trained to learn lower dimension data representations at the bottleneck of the network. They have the capability to learn representations in a fully unsupervised way which makes them suitable for the task at hand. The encoder($g_{V_E}(.)$) consists of the input(dim=$m$), hidden and the output layers(dim=$d$). The decoder($g_{V_D}(.)$) uses this low dimension representation to reconstruct the original input data. The decoder is a mirrored version of the encoder. The input layer is fully connected to the output layers with an intermediate ReLU activation layer in both the encoder and the decoder. We initialize the network weights using kaiming normal initialization. The output from the encoder is passed through a softmax layer to generate a soft version of deep $K$-means [Fard et al., 2020] which gives the assignment simplex $\tilde{a}(\psi) \sim \bar{\pi}^\theta(g_{V_E}(\psi))$ where

$$\bar{\pi}_k^\theta(g_{V_E}(\psi)) := \frac{\exp\{-\beta\|g_{V_E}(\psi) - \theta^k\|^2\}}{\sum_{k'=1}^K \exp\{-\beta\|g_{V_E}(\psi) - \theta^{k'}\|^2\}} \tag{18}$$

The parallel network($\{f_{W^k}\}_{k=1}^K$) designs the $K$ customized data-driven uncertainty sets using a slightly modified deep SVDD method from [Goerigk and Kurtz, 2020]. The input to these networks is the perturbations $\xi$ and the assignment policy($\bar{\pi}^\theta(g_{V_E}(\psi))$). Each $f_{W^k}$ has an input layer(dim=15), hidden layer and an output layer(dim=5). All layers are fully connected with a ReLU activation function. All the networks are initialized with a uniform distribution in $[0, 1]$. Our approach constructs a weighted center, $\bar{f}_{W^k|\tilde{a}(\psi_i)=k}^{\theta,V}$ which uses $\bar{\pi}^\theta(g_{V_E}(\psi))$ to compute the loss (17).

## B.3 Suggested extensive parameter tuning procedure

In this section, we discuss the parameter tuning strategy that can be used to train the network proposed in section B.2 using the portfolio optimization example discussed in section 5.2. Here, given the time series nature of the data, we follow the rolling window approach for network training. Our architecture uses a set of hyperparameters, $hp = (lr, \alpha_K, \alpha_S, \beta, K)$ where $lr$ represents the learning rate, $\alpha_K$ regulates the trade-off between seeking good representations for $\psi$ that are faithful to the original data and representations that are useful for clustering purposes. $\alpha_S$ plays a similar trade-off between the recognizability and compactness of uncertainty sets. Finally, $\beta$ is a softmax temperature parameter and $K$ represents the number of clusters. We split the data into training and validation periods and search for the optimal combination through the grid search method. For each combination, we train the network and generate the optimal policy using training data which is applied to the unseen validation data. The optimal combination is the one that gives the lowest $VaR_{1-\epsilon}$ on the validation dataset as this is a worst case return minimization problem. This is shown in algorithm 2. Once the hyperparameters are selected, we re-train the network using the complete data. It is important to note that the results reported in section 5 did not use parameter tuning to reduce computations.

## B.4 Simulated data generation process

In this section, we discuss the data generation process for the simulated data used in section 5.1. For easy visualization, we consider a simulation environment where $[\psi^T \ \xi^T]^T \in \mathbb{R}^4$ is a random vector whose distribution is an equal-weighted mixture of two 4-d multivariate normal distributions.[7]

---

[6]Here, $\mathcal{S}^k$ refers to $(\bar{f}_{W^k|\tilde{a}(\psi_i)=k}^{\theta,V}, \bar{\Sigma}_{W_k|\tilde{a}(\psi)=k}^{\theta,V})$ with

$$\bar{\Sigma}_{W_k|\tilde{a}(\psi)=k}^{\theta,V} := \sum_{i=1}^N \frac{\bar{\pi}_k^\theta(g_{V_E}(\psi_i))}{\sum_{i=1}^N \bar{\pi}_k^\theta(g_{V_E}(\psi_i))} \cdot (f_{W^k}(\xi_i) - \bar{f}_{W^k|\tilde{a}(\psi_i)=k}^{\theta,V})(f_{W^k}(\xi_i) - \bar{f}_{W^k|\tilde{a}(\psi_i)=k}^{\theta,V})^T,$$

.

[7]The data is generated using [Page Jr, 1984].

**Algorithm 2** Hyperparameter tuning

---

Input:$hp = (lr, \alpha_K, \alpha_S, \beta, K)$
**for** $year = y, \ldots, y + M$ **do**
    Obtain $\{\mathcal{U}^k\}_{k=1}^K$ from Algorithm 1
    Get optimal portfolio using: $\min_{x \in \mathcal{X}} \text{VaR}_{1-\epsilon}(\xi^\mathsf{T} x)5.2$
**end for**
Choose $hp$ which minimizes out of sample $VaR_{1-\epsilon}$ for $M$ periods

---

Namely, $[\psi^T \ \xi^T]^T \sim 0.5N(\mu_1, \Sigma_1) + 0.5N(\mu_2, \Sigma_2)$ where:

$$
\mu_1 := \begin{bmatrix} 1 \\ 2 \\ 0 \\ 4 \end{bmatrix}, \qquad
\Sigma_1 := \begin{bmatrix} 1.0 & 0.0 & 0.3 & -0.1 \\ 0.0 & 1.0 & 0.1 & -0.2 \\ 0.3 & 0.1 & 1.0 & 2.0 \\ -0.1 & -0.2 & 2.0 & 1.0 \end{bmatrix}
$$

$$
\mu_2 := \begin{bmatrix} 5 \\ 5 \\ 4 \\ 0 \end{bmatrix}, \qquad
\Sigma_2 := \begin{bmatrix} 1.0 & 0.0 & 0.3 & -0.1 \\ 0.0 & 1.0 & 0.1 & -0.2 \\ 0.3 & 0.1 & 1.0 & 0.0 \\ -0.1 & -0.2 & 0.0 & 1.0 \end{bmatrix}.
$$

The distribution marginalized over the random vectors $\psi \in \mathbb{R}^2$ and $\xi \in \mathbb{R}^2$ can respectively be visualized in figure 3(a) and (b).

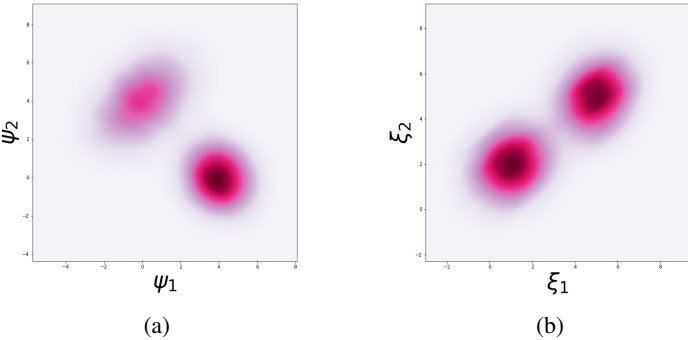

(a)                            (b)

Figure 3: Density plot of the marginalized distributions over $\psi$ (in (a)) and $\xi$ (in (b)) from a mixture of two Gaussian distributions on the joint space $[\psi^T \ \xi^T]^T$.

## B.5 Sensitivity analysis for parameters

Here, we show the sensitivity analysis for the parameters $\alpha_S$ and $K$. For each of these analyses, we keep all the other parameters constant and train the model by varying the considered parameters. For $\alpha_S$, we consider the range of values between 0 and 1. For the sensitivity analysis of $K$, we considered 1 to 9 clusters. We conducted 10 such runs in the year 2019 and observe the average validation VaR. The results can be seen in the plots below. The analysis in 4b shows that 2 clusters result in similar or improved performance compared to using more clusters. Regarding the influence of $\alpha_S$ on out-of-sample performance, we did not observe any insightful behavior. We believe this hyperparameter can play a role in problem settings where the convergence of TV losses in contextual and perturbed spaces is different and needs moderation. However, in this case, we don't notice any such issues and the choice of $\alpha_S$ as 0.5 seemed to work generally well across all experiments as seen in 4a. The sensitivity analysis also highlights the same, which points to 0.5 as being a legitimate choice for $\alpha_S$.

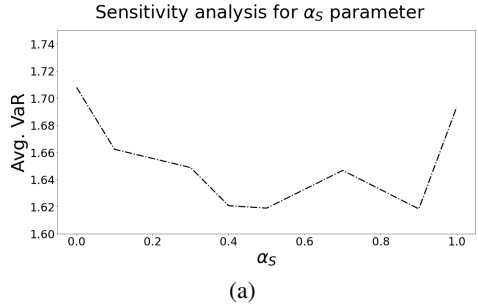
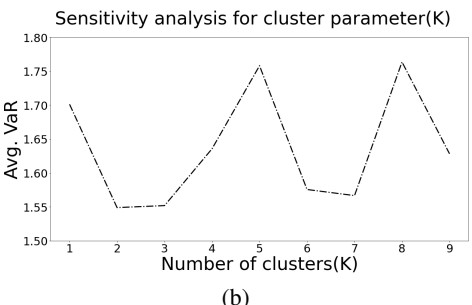

(a)                                        (b)

Figure 4: Sensitivity analysis (using validation data) across portfolio simulations for the year 2019.

# C    Algorithms

In this section, we provide the pseudo-code for the Iterative constraint generation and the Deep Cluster then Classify 4.1 techniques.

## C.1    Iterative constraint generation

We present the iterative constraint generation algorithm for both the robust objective problem:

$$\min_{x \in \mathcal{X}} \max_{\xi \in \mathcal{U}} c(x, \xi),$$

and a robust constraint problem of the form:

$$\min_{x \in \mathcal{X}: c(x,\xi) \leq 0, \forall \xi \in \mathcal{U}} f(x).$$

We note that when $\mathcal{X}$ is convex and $c(x,\xi)$ is convex in $x$ and linear in $\xi$, then $\arg\min_{x \in \mathcal{X}} \max_{\xi \in \mathcal{U}'} c(x,\xi)$ can be obtained using convex optimization algorithms, while $\xi^* \in \operatorname{argmax}_{\xi \in \mathcal{U}} c(x^*, \xi)$ can be obtained using mixed-integer linear programming solvers such as MOSEK (see MOSEK ApS [2022]). In more general setting, one might need to employ more general non-linear programming software.

---

**Algorithm 3** Iterative constraint generation for robust objective problem

---

Input: Max number of iteration $M$
Set $\mathcal{U}' := \{\xi_0\} \subseteq \mathcal{U}$
**for** $iter = 1, \dots, M$ **do**
    Set $x^* \in \arg\min_{x \in \mathcal{X}} \max_{\xi \in \mathcal{U}'} c(x, \xi)$
    Set $\xi^* \in \operatorname{argmax}_{\xi \in \mathcal{U}} c(x^*, \xi)$
    **if** $c(x^*, \xi^*) > \max_{\xi \in \mathcal{U}'} c(x^*, \xi)$ **then**
        Add $\xi^*$ to $\mathcal{U}'$
    **else**
        Break
    **end if**
**end for**
Return $x^*$

---

---

**Algorithm 4** Iterative constraint generation for robust constraint problem

---

Input: Max number of iteration $M$
Set $\mathcal{U}' := \{\xi_0\} \subseteq \mathcal{U}$
**for** $iter = 1, \ldots, M$ **do**
    Set $x^* \in \arg\min_{x \in \mathcal{X}: c(x,\xi) \leq 0, \forall \xi \in \mathcal{U}'} f(x, \xi)$
    Set $\xi^* \in \arg\max_{\xi \in \mathcal{U}} c(x^*, \xi)$
    **if** $c(x^*, \xi^*) > 0$ **then**
        Add $\xi^*$ to $\mathcal{U}'$
    **else**
        Break
    **end if**
**end for**
Return $x^*$

---

## C.2   Deep Cluster then Classify with deep $K$-means

---

**Algorithm 5** Deep Cluster then Classify with deep $K$-means

---

**Input:** Data-set $\mathcal{D}_{\psi\xi}$, number of clusters $K$, maximum number of iterations $T$, coverage error $\epsilon$
Randomly initialize $\theta_0$, $V_0$ and all $W_0^k$'s
Let $a_0(\psi) := \bar{a}^{\theta_0}(g_{V_{E_0}}(\psi))$
Set $t := 0$
**repeat**
    Set $t := t + 1$.
    Update $\theta_t^k := \sum_{i \in \mathcal{I}_k} g_{V_{E_{t-1}}}(\psi_i)/|\mathcal{I}_k|$ where $\mathcal{I}_k := \{i : a_{t-1}(\psi_i) = k\}$
    Let $a_t(\psi) := \bar{a}^{\theta_t}(g_{V_{E_{t-1}}}(\psi))$
    Update $V_t$ using SGD on (8) with $a_t(\psi_i)$
**until** $t \geq T$
Let $a(\psi) := a_t(\psi)$
**for** $k = 1, \ldots, K$ **do**
    Train the parameters $W^k$ using (5) with $\mathcal{D}_\xi^k$
    Calibrate $R^k$ on $\mathcal{D}_\xi^k$ using coverage target $1 - \epsilon$
    Let $\mathcal{U}^k := \mathcal{U}(W^k, R^k, \mathcal{S}^k)$
**end for**
Return $a(\cdot)$ and $\{\mathcal{U}^k\}_{k=1}^K$

---