# OpenReview forum: "Data-Driven Conditional Robust Optimization"
_NeurIPS.cc/2022/Conference — NeurIPS 2022 Accept_

### Official Review · Reviewer_34vX · 2022-07-11

**Rating:** 7
**Confidence:** 5
**Soundness:** 3 good
**Presentation:** 3 good
**Contribution:** 3 good

**Summary:**

In this paper, the author has proposed a Conditional Robust Optimization (CRO) paradigm that seeks the solution to a robust optimization problem using contextual information. The uncertainty set accounts for the most recent side information provided by the set of covariates.
The conditional uncertainty is designed with an integrated framework (IDCC) that jointly learns a partition in the covariate data space and simultaneously constructing region specific deep uncertainty sets for the random vector that perturbs the CRO problem. The paper tested the
IDCC algorithm using the US stock market dataset, and shows empirically that their approach reduces the out-of-sample VaR considerably compared to non-contextual RO schemes when the level of protection needed is high.

**Questions:**

1. Use of more real or synthetic datasets to get a better generalization of the approach in various settings, and to clarify some of the claims, such as the fact that the loss in performance for minority clusters is due to its low frequency which makes it difficult to
learn the conditional uncertainty sets.
2. It might be nice to have the algorithms for the DCC also mentioned to help us understand the differences between DCC and IDCC, as well as the individual approaches, more intuitively.
3. Here the author has used K-means clustering, which works very well for smaller data, but can take very long to run for large datasets, since it iterates over the entire dataset. It can be useful to try out larger datasets using density based clustering methods such as
DBSCAN. Further, it might be useful to try out some soft clustering methods like fuzzy k-means or fuzzy c-means for partial clustering.
4. A mention of the run-time of each of the experiments might give a better idea of the comparisons of the various methods, unless their run-times are similar.
5. Are the above algorithms useful only for time-series data? Can we extend these algorithms to perhaps a computer vision system? If yes, it would be nice to generalize the approach in various different settings either through a short description or more
variety of datasets.
6. Can the authors mention more explicitly the limitations of the proposed algorithm?

**Limitations:**


Tied to K-Means Clustering although it's clear now that the clustering algorithm can be tailored.

**Strengths And Weaknesses:**

1. CRO is a novel approach that utilizes the contextual information. It reduces the out of sample value at risk considerably compared to the conventional non-contextual optimization methods.
2. The paper corrects a wrong assumption made previously by the authors of the DDDRO approach that the projected perturbation vector is normalized for each data, by replacing the simply defined uncertainty set that uses an ellipsoid in the projected space instead of
a sphere.
3. The IDCC framework performs deep clustering and deep uncertainty set design jointly which ensures that the uncertainty set is learnt correctly, by ensuring that low variation in the side information vector results in low variation in the perturbation vector.
4. Deals well with partial clustering situations by training a parameterized random assignment policy.
5. DDDRO and IDCC, which use deep neural networks, are capable of identifying non-convex uncertainty sets.
6. IDCC exploits the side information by concentrating the uncertainty set on a region that has most mass thus leading to a less conservative optimization problem than DDDRO and ellipsoid.
7. The paper is well documented in terms of the details for the experiments with no significant ambiguities.

---

> ### Author Response · Authors · 2022-08-02
> **Addressing reviewer's feedback**
>
> Q1. We understand the point of view that it would be interesting to evaluate the approach on other synthetic or real data sets. However, we wish to emphasize that the portfolio optimization problem is a classical problem to investigate in a risk-averse setting (given that it becomes a trivial problem when returns are pre-determined) and known to be highly challenging when deployed using real market data. This difficulty is due to the amount of noise in the data, its non-stationarity, and the fact that the financial market is typically arbitrage-free (implying that a higher expected return comes at the price of higher risk). We would like to point out that we did experiment with the synthetic data as well but decided to only present the insights about the structure of uncertainty sets as the out-of-sample performances were very similar to what we see in the portfolio optimization problem. Moreover, the portfolio problem has been solved across three different time periods and 10 different sets of assets to demonstrate the robustness of the approach.
>
> Q2. We direct the reviewer to algorithm 5 (in the updated manuscript, previously algorithm 4) in the appendix for the DCC algorithm. The key difference between the DCC and IDCC is that the DCC approach constructs the uncertainty sets in a decoupled way whereas IDCC does it by jointly minimizing the total variance in both contextual feature and perturbation vector space. We direct the reviewer to the beginning of section 4.2 for more information about the differences between the approaches and how IDCC improves over the DCC approach.
>
> Q3. We thank the reviewer for the suggestions. We do agree that K-means clustering might run into issues for larger data sets but for the problem considered, we did not notice any such issues. This being said, our conceptual approach is general and flexible. We decided to exploit a loss function that employs total variation which further motivated us in using a soft K-means approach. With suitable modification to make the output a soft clustering, one could incorporate alternative clustering techniques like fuzzy k-means or DBSCAN into our approach.
>
>
> Q4. We appreciate this insight from the reviewer. The DDDRO, DCC, and IDCC approaches have very similar experiment run times (~ 10 min for 4 different \epsilon ) and the ellipsoid method (~ 2 min) is faster than those methods. The point to note is that the ellipsoid method doesn’t involve training any neural network components which is the differentiating factor.
>
> Q5. We would like to highlight that none of the approaches discussed in the paper restrict the DM to time series data and all can be used whenever suitable datasets are available. In fact, the side information can take any form, be it structured or unstructured. We present some more applications below to highlight the diverse applications possible.
>
> In the portfolio optimization problem, we use various market indices as side information. Along with this, we could use text data from social media as a proxy for the sentiment for the stock as side information as well. This text data can be converted to embeddings and can be seamlessly integrated into our approach.
> One could also use our approach to solve a conditional shortest path problem, which consists in finding the fastest route between two endpoints in a city conditional on information about weather, traffic conditions, etc. One could use computer vision systems to gather richer side information like the traffic density, and road conditions which can be used in real-time to produce robust routing plans.
> We hope that these applications provide the necessary motivation to view our algorithm as a generalized approach to solving CRO problems.
>
> Q6. All the methods discussed in the previous literature construct uncertainty sets from historical observations whereas IDCC adapts them to contextual information. To do this, we use a clustering-based approach. This method has the advantage of generalizing across different conditions. However, the notion of clustering can be a limitation and we focus our future research to develop sets that are customized to each realized vector of features. Moreover, the current methodology is limited to single-stage RO problems and we leave it to the future score to extend it to multi-stage CRO applications.
>
> Limitations:
> We would once again like to direct the reviewer to our response for Q3 and Q5. In fact, we disagree with the reviewer regarding the two identified limitations. Indeed, our model is in no way limited to time series datasets (our implementation did not exploit this structure of the dataset) and is flexible enough to integrate various types of clustering techniques. We would be happy to discuss further any other issues that the reviewer might have observed.

---

> ### Comment · Reviewer_34vX · 2022-08-08
> **Authors' response**
>
> I'm happy with the explanation and the fact that the method is amenable to different clustering techniques. I have raised my score accordingly.

---

### Official Review · Reviewer_8P9p · 2022-07-11

**Rating:** 7
**Confidence:** 2
**Soundness:** 4 excellent
**Presentation:** 3 good
**Contribution:** 4 excellent

**Summary:**

This paper addresses the robust optimization problem with the covariates provided and provides data-driven solutions. It proposes an integrated framework for performing deep clustering and deep uncertainty set design jointly to ensure the total variation in the covariate space can provide a similar guarantee for the one in the perturbation parameters. Moreover, they rigorously prove the connections between conditional robust optimization and the CVO problem. Simulation data and robust portfolio optimization using real-world data from the US market are provided for empirical evidence.

**Questions:**

None.

**Limitations:**

Cannot find any limitations about this paper.

**Strengths And Weaknesses:**

This paper is the first to introduce the conditional robust optimization framework for solving contextual optimization problems in a risk-averse setting based on the classical robust optimization approach but with novel development, i.e., an integrated approach to design the uncertainty sets adapted to the covariate information. The story is complete and the analysis is comprehensive. The problem is well-motivated and the solutions are sound and concrete.

---

> ### Author Response · Authors · 2022-08-02
> **Thanking the reviewer for the feedback**
>
> We thank the reviewer for their positive feedback. We are however surprised to see a “Weak accept” recommendation given that the reviewer agrees that we are the first to propose this framework and that our development is novel. We would be happy to discuss any issues they have identified that prevent the paper from having a higher impact on the NeurIPS community. Note that our work integrates ML techniques from clustering, one-class classification (i.e. outlier detection), and optimization in a novel and original way. We believe it paves the way for many other research initiatives at this intersection.

---

> > ### Comment · Reviewer_8P9p · 2022-08-08
> > **Thanks for the authors responses**
> >
> > I am not an expert in this specific field but I read the other reviewers' comments as well as your responses. From your extra explanations and extra experimental results, I am now more confidence to say that you address the problem in a sound way and tell a completed story. I will raise my scores accordingly.

---

### Official Review · Reviewer_k1N2 · 2022-07-11

**Rating:** 8
**Confidence:** 4
**Soundness:** 3 good
**Presentation:** 3 good
**Contribution:** 4 excellent

**Summary:**

The authors study risk aversion in a data-driven problem with contextual data using robust optimization. They extend previous work that uses deep neural networks to build unconditional uncertainty sets. They develop conditional uncertainty sets that leverage contextual data. Building conditional uncertainty sets for a new observation is based on two steps: (1) assigning the new observation to a data cluster, and (2) creating a local unconditional uncertainty set for the cluster. The authors present two methods to build these clusters based on minimizing total variation. The second method is a generalization of the first method and allows a trade-off minimizing variations in the feature space and disturbance space.

The performance of the method is demonstrated on a portfolio optimization problem. The results show that the integrated data-driven approach (IDDC) provides the best risk reduction for high-risk aversion, and when data can be clustered in two clusters that are relatively balanced.

**Questions:**

Exposition:

(Q1.1) What is the intuition/motivation for minimizing total variations of features/disturbances when building uncertainty sets? in particular: why should this lead to more robust decisions than elliptical or polyhedral uncertainty sets?

(Q1.2) The approach seems limited to linear uncertainty in the cost function. Could this be extended to a cost function that is non-linear in the uncertainty (this happens already for simple problems e.g. newsvendor with uncertain demand) or to uncertainty in the constraints?

(Q1.3) Is my understanding correct that the uncertainty sets are "constant" within each cluster? Could this be improved? There seems to be a trade-off between having many clusters using few data points but being more adaptive, and using few clusters but maybe not using the contextual data to its fullest extent. I think a sensitivity analysis of the number of clusters and its impact on decisions and costs could be interesting.

(Q1.4) I do not understand the statement regarding the "deﬁciency of DDDRO approach, who assume wrongfully that the projected fWk(ξ) are normalized" (p.4, lines 161-162). Could you explain it in more detail?

(Q1.5) Could you develop the intuition behind the first drawback of the DCC approach? (p. 5, lines 171-121) This is probably related to (Q1.1).

(Q1.6) The statement "as α_S → 1, we can expect the minimizer of L_α^3 to converge to the minimizer of the cluster then classify approach" seems unnecessarily vague (p.5, line 194). It seems that α_S = 1 reduces exactly IDCC to DCC. Is this correct? I would also be interested in seeing a sensitivity analysis of α_S and how it impacts decisions and costs.

(Q1.7) The cluster assignment is a random policy. Is this a problem in practice? Random decisions may be hard to use and accept by practitioners. Why not use the most likely cluster?

Numerical study:

(Q2.1) Why is there no benchmark from recent literature outside of DDDRO on which the work is based? Comparing with a data-driven benchmark without risk consideration (e.g. Bertsimas & Kallus (2019) or Bertsimas et al. (2022) who also use ideas from robust optimization) and for instance with a distributionally robust benchmark without conditional information could provide a more precise idea of the value of the approach. Bertsimas & Kallus (2019) could be an easily implementable benchmark: it uses a Machine Learning model to weight historical data and provides a conditional empirical probability distribution. The (conditional) value-at-risk portfolio optimization model could then be optimized using a sample-average method. This approach has been discussed by Lin et al. (2022) for a risk-averse newsvendor problem.

(Q2.2) Is the number of replications sufficient (N=10) to highlight a statistical difference between the performance of the methods (especially since IDCC is a random policy)? What do the error bars in Figure 2 show (confidence interval, standard deviation)?

(Q2.3) What are the majority and minority clusters? This suggests that only K=2 clusters are used, but it is said on line 269 that the number of clusters is optimized.

(Q2.4) Is \epsilon used to build the uncertainty sets radii the same as when evaluating VaR_\epsilon? If (I)DCC is a conservative approximation of Var(\epsilon) then maybe the best value of \epsilon to build the sets is smaller than the actual target. I would also be interested in seeing the effect of \epsilon on the trade-off between average costs and value-at-risk.

Minor comments:
- I cannot find the footnote relating to note [1] on page 1.
- I believe \alpha_K and \alpha_S are not properly introduced.
- I believe that when Algorithms 2 and 3 are referenced in the main paper, it should be Algorithms 3 and 4 resp.

References:
Dimitris Bertsimas and Nathan Kallus. From predictive to prescriptive analytics. Management Science, 66(3):1025–1044, 2020.
Dimitris Bertsimas, Christopher McCord, and Bradley Sturt. Dynamic optimization with side information. European Journal of Operational Research, 2022.
Lin, Shaochong, Youhua Chen, Yanzhi Li, and Zuo‐Jun Max Shen. Data‐Driven Newsvendor Problems Regularized by a Profit Risk Constraint. Production and Operations Management 31(4):1630-1644, 2022.

**Limitations:**

I find the work to be very complete. I see only two significant limitations that are already detailed in the question part: (1) the lack of benchmark and comparison to recent literature, and (2) not discussing how to extend the approach to problems where the cost function is not linear in the uncertainty.

**Strengths And Weaknesses:**

The paper provides an interesting approach to a well-motivated and significant problem. I find the methodology sound and well presented. In particular, the approach leverages very well results from diverse fields such as clustering, outlier detection, robust optimization, and mixed-integer linear optimization.
My main concern relates to the experimental setting, in particular the lack of benchmark from recent and relevant literature. I also have several questions/suggestions regarding the exposition of the method.

---

> ### Author Response · Authors · 2022-08-02
> **Clarification of reviewer's remarks and additional numerical analysis**
>
> 1.1&1.5 On the feature side, using total variation promotes projections that create clusters that are easier to distinguish from each other as TV is smaller within than across clusters. On the disturbance side, it promotes projections concentrated around their mean, so that the inverse projection of the ellipsoid tightly contains the data points. The latter can lead to better robust decisions, not in terms of robustness(all offer the same protection) but rather in terms of not protecting against more scenarios than needed.
>
> 1.2 None of our results assume a linear cost function. For non-linear case, some challenges might arise when seeking the worst-case scenario in algorithm 3, where mixed-integer non-linear programming methods are then needed (Belotti et al.). These are typical issues in the application of RO. The new algorithm 4 has the variation for robust constraints.
> Belotti et al. (2013). Mixed-integer nonlinear optimization. Acta Numerica
>
> 1.3 The uncertainty sets are indeed constant within a cluster. Future research should investigate sets customized to each realized vector of features. Yet, there is indeed an inherent risk that the amount of adaptation in the conditional set not be justified by the size of data. We now present a sensitivity analysis (new appendix B.5) that motivates the choice K=2.
>
> 1.4 “Normalized” refers to the variance of each term of the projected disturbance being the same. Goerigk&Kurtz wrongfully assumes that the cov. matrix of the projected data points is a scaled identity matrix.
>
> 1.6 The statement “α_S =1 reduces exactly IDCC to DCC” is not accurate given that when α_S =1 the loss function will not orient the choice of W^k’s anymore. Furthermore, we emphasize that IDCC employs randomized cluster assignments while DCC does not. Regarding the influence of α_S, we did not observe any insightful behavior. We now include in appendix B.5 the requested sensitivity analysis, which points to 0.5 as being a legitimate choice for α_S.
>
> 1.7 We also experimented with most likely clusters and the performance remained very similar. Note however that Lemma 4.1 and 4.2 do not necessarily hold anymore when implemented this way. The random policy might be easier to adopt if DM are only shown the uncertainty set associated with the randomly drawn cluster, which would resemble the use of statistical confidence intervals, namely that in the long run these cover the true realization 1- ϵ proportion of the time.
>
> 2.1 Our focus is CRO rather than CSO. RO is a rich modeling paradigm on its own that circumvents the need to think in terms of an underlying distribution model. Uncertainty sets can also be more easily statistically validated(e.g. χ2 test) compared to stochastic models or distribution sets.
> When one uses CSO to produce a CRO set, they still face the question of the right “shape” for this set, which is the focus of DDRO. We therefore claim that our current experiments offer a legitimate proof of concept for our new CRO technique. This being said, we believe there is space to investigate many potential synergies between our work and CSO, e.g one could apply the non-parametric CSO methods in the projected space $g_{V_E}(\psi)$ instead of $\psi$.
> Finally, based on the suggestion from the reviewer, we implemented a preliminary version of Bertsimas & Kallus(2019) that uses KNN and compared its performance in 2019. The results are shown in the table below, IDCC dominates(with high confidence) across the quantiles.
> |       | 0.8  | 0.9  | 0.95 | 0.99 |
> |-------|------|------|------|------|
> | IDCC  | 0.44 | 0.77 | 1.11 | 2.02 |
> | DDDRO | 0.45 | 0.84 | 1.27 | 2.35 |
> | DCC   | 0.47 | 0.81 |  1.3 | 2.52 |
> | CSO   | 0.58 | 1.05 | 1.54 | 2.12 |
>
> 2.2 The error bars represent the 95% CI. With N=10, random policies are applied 2500 times. We conducted statistical significance tests and found that in 5 out of the 6 cases IDCC performs better(out-of-sample VaR) than both DDDRO and ellipsoid uncertainty set with 90% confidence when we focus on the 95 and 99 quantile levels.
>
> 2.3 The maj./min. represents the frequency of the occurrence of the clusters. As in table 1, this helped to classify the test data into clusters with different risk profiles.
> In this experiment, we used K=2 which gave good results based on preliminary studies. The motivation for K=2 can also be obtained from the new sensitivity analysis now presented in Appendix B.5.
>
> 2.4 The reviewer is right that in some special cases a larger ϵ might lead to a more accurate representation of the VaR(e.g.if ξ is multivariate normal in our application). The paper is however focused on RO rather than VaR minimization. Hence, our experiments require the methods to construct uncertainty sets with a certain coverage probability.
>
> Minor comments:
> We thank the reviewer for pointing out these issues, we have removed the footnote and added some context about α_k and α_S to introduce them better. Referencing issues with the algorithms are also rectified

---

> > ### Comment · Reviewer_k1N2 · 2022-08-07
> > **Follow-up**
> >
> > I thank the authors for their detailed response. All my questions and limitations have been addressed, and I find the additional experiments valuable. Therefore, I increased my score accordingly.

---

### Author Response · Authors · 2022-08-02
**Thanking the reviewer's for the feedback**

We would like to thank the reviewers for their thoughtful comments and their efforts toward improving the manuscript. In the following, We address their comments/questions.  Minor additional details following the reviewer's suggestions are added to the manuscript and they appear in blue.

---

### Meta-Review · Area_Chair_Kt86 · 2022-08-26

**Recommendation:** Accept
**Confidence:** Certain

**Metareview:**

The paper proposes a conditional robust optimization framework for solving contextual optimization problems in a risk-averse setting. All three reviewers seem to agree on the usefulness and originality of the proposed approach. As Reviewer 8P9p finds the story to be complete, the analysis to be comprehensive and the problem to be well-motivated with sound and concrete solutions, Reviewer k1N2 points out that the approach leverages very well results from diverse fields such as clustering, outlier detection, robust optimization, and mixed-integer linear optimization. Reviewer 34vX stresses that the method reduces the out of sample value at risk considerably compared to the conventional non-contextual optimization methods. While k1N2 originally phrased some concerns relating to the experimental setting, in particular the lack of benchmark from recent and relevant literature, the autor response lead k1N2 to a score increase. Similarly, 8P9p stated following the author response that from their extra explanations and extra experimental results, the reviewer gained confidence in the soundness of the proposed approach, also resulting in a score increase. Reviewer 34vX put to the fore that the experimental aspects of the paper are well documented. Also, the IDCC algorithm of the paper is illustrated using the US stock market dataset, showing empirically that the approach reduces the out-of-sample VaR considerably compared to non-contextual RO schemes when the level of protection needed is high. From a methodological perspective, it is also point out by 34vX that as neural network-based approaches, the considered DDRO and IDCC approaches are capable of identifying non-convex uncertainty sets, a desirable property. Overall, this paper was found to be a very well written and original reseach work on a timely topic w,ith methodological and empirical results of potentially string interest to the NeurIPS readership and beyond.  For all these reasons, I am recommending the paper to be accepted.

**Award:**

Yes

---

### Decision · Program_Chairs · 2022-09-14

Accept